# Byzantine Spectral Ranking

**Arnhav Datar** *
Indian Institute of Technology, Madras
adatar@cmu.edu

**Arun Rajkumar** *
Indian Institute of Technology, Madras
arunr@cse.iitm.ac.in

**John Augustine**†
Indian Institute of Technology, Madras
augustine@iitm.ac.in

## Abstract

We study the problem of rank aggregation where the goal is to obtain a global ranking by aggregating pair-wise comparisons of voters over a set of objects. We consider an adversarial setting where the voters are partitioned into two sets. The first set votes in a stochastic manner according to the popular score-based Bradley-Terry-Luce (BTL) model for pairwise comparisons. The second set comprises malicious *Byzantine* voters trying to deteriorate the ranking. We consider a strongly-adversarial scenario where the Byzantine voters know the BTL scores, the votes of the *good* voters, the algorithm, and can collude with each other. We first show that the popular spectral ranking based Rank-Centrality algorithm, though optimal for the BTL model, does not perform well even when a small constant fraction of the voters are Byzantine.

We introduce the Byzantine Spectral Ranking Algorithm (and a faster variant of it), which produces a reliable ranking when the number of good voters exceeds the number of Byzantine voters. We show that no algorithm can produce a satisfactory ranking with probability $> 1/2$ for all BTL weights when there are more Byzantine voters than good voters, showing that our algorithm works for all possible population fractions. We support our theoretical results with experimental results on synthetic and real datasets to demonstrate the failure of the Rank-Centrality algorithm under several adversarial scenarios and how the proposed Byzantine Spectral Ranking algorithm is robust in obtaining good rankings.

## 1 Introduction

Rank aggregation is a fundamental task in a wide spectrum of learning and social contexts such as social choice [Soufiani et al., 2014, Caplin and Nalebuff, 1991], web search and information retrieval [Brin and Page, 1998, Dwork et al., 2001], recommendation systems [Baltrunas et al., 2010] and crowd sourcing [Chen et al., 2013]. Given pairwise comparison information over a set of $n$ objects, the goal is to identify a ranking that best respects the revealed preferences. Frequently, we are also interested in the scores associated with the objects to deduce the intensity of the resulting preference. There have been several solutions to this problem including Spectral Ranking methods such as Rank-Centrality [Negahban et al., 2017] and MLE methods [Ford, 1957].

The Bradley-Terry-Luce (BTL) model [Bradley and Terry, 1952, Luce, 1959] has been prevalent for a vast variety of ranking algorithms. The BTL model assumes that there are $n$ objects that are to be

---

*Supported in part by the Robert Bosch Centre for Data Science and Artificial Intelligence, IIT Madras.

†Supported in part by the Cryptography, Cybersecurity, and Distributed Trust pCoE and the Accenture CoE at IIT Madras.

compared and each of these objects has a positive weight ($w_i$) associated with itself. Whenever any voter is asked a query for a pair, the voter claims $i$ is better than $j$ with probability $w_i/(w_i + w_j)$ independently at random. However, the BTL model assumes that all voters are identical and use the same probabilistic strategy for voting. Generally, in crowd-sourced settings, voters differ significantly from each other. Some of the voters may be spammers or even direct competitors of the entity ranking the objects, leading to data poisoning [Sun et al., 2021]. To model this situation, we assume that there is a split in the population. We consider that there are $K - F$ *good* voters and $F$ *Byzantine* voters, to appropriately model the division of the voters. The good voters vote according to the BTL model, while the Byzantine voters can potentially be controlled by an adversary that can centrally coordinate their behavior.

## 1.1 Overview of our Results

We naturally initially consider the tolerance of the Rank-Centrality algorithm against a constant fraction of Byzantine voters. Negahban et al. [2017] showed that for an Erdős-Rényi pairwise-comparison graph $G(n, p)$ we have

**Theorem 1** (Informal from Negahban et al. [2017]). *If $p \in \Omega(\log n/n)$ and $k \in \Omega(1)$ then the Rank-Centrality algorithm outputs a weight-distribution ($\pi$) such that with high probability*[3]

$$\frac{\|\pi - \tilde{\pi}\|}{\|\tilde{\pi}\|} \in \mathcal{O}\left(\sqrt{\frac{\log n}{knp}}\right)$$

Here $\tilde{\pi}$ are the true weights of the objects and $k$ is the number of voter queries asked per edge in $G$. We can see that when $k \in \omega(1)$ the RHS goes to 0 as $n$ goes to $\infty$. We show that a strategy as simple as all Byzantine voters voting the reverse of the weights causes the Rank-Centrality algorithm to output weights far from true weights with high probability.

**Theorem 2** (Informal). *There exists a strategy that the Byzantine voters can use such that the Rank-Centrality algorithm outputs a distribution $\pi$ such that with high probability*

$$\frac{\|\pi - \tilde{\pi}\|}{\|\tilde{\pi}\|} \in \Omega(1)$$

*provided there are a constant fraction of Byzantine voters*

Even simple algorithms that output the same weights regardless of the voter's responses also give $\Theta(1)$ error. To tackle this situation, we propose Byzantine Spectral Ranking, an algorithm that runs in polynomial time to remove some of the voters such that the Rank-Centrality algorithm can be applied. As long as the proportion of Byzantine voters is bounded strictly below $1/2$, the algorithm removes some voters such that the remaining Byzantine voters are unlikely to cause the Rank-Centrality algorithm to deviate significantly.

**Theorem 3** (Informal). *For a range of values of $k$, there exists an $\mathcal{O}(n^2)$-time algorithm that removes voters such that with high probability we have:*

$$\frac{\|\pi - \tilde{\pi}\|}{\|\tilde{\pi}\|} \in \mathcal{O}\left(\frac{d_{max}}{k}\right)$$

Here $d_{max}$ is the maximum degree of the comparison graph. Finally, we also show that when $F \geq K/2$ there is no algorithm that can solve this problem for all weights with probability $> 1/2$, by coming up with a strategy that the Byzantine voters can use for two different weight distributions, and showing therefore that the algorithm must fail in at least one of the weight distributions.

**Theorem 4** (Informal). *If $F \geq K/2$, then there is no algorithm that can for all weights ($\tilde{\pi}$) with probability $> 0.5$, give a weight distribution ($\pi^*$) such that*

$$\frac{\|\pi^* - \tilde{\pi}\|}{\|\tilde{\pi}\|} \in o(1)$$

---

[3]Whenever we say with high probability we mean with probability $\geq 1 - \frac{1}{n^c}$ for $c \geq 0$

## 1.2 Related Works

In recent times, there has been a lot of work on building robust machine learning algorithms [Blanchard et al., 2017, El Mhamdi et al., 2018, Li et al., 2020, El-Mhamdi et al., 2021, Chen et al., 2018, Wu et al., 2021] that are resilient to Byzantine faults.

Ranking from pairwise comparisons has been a very active field of research and there have been multiple algorithms proposed with regard to the BTL model [Negahban et al., 2017, Agarwal et al., 2018, Rajkumar and Agarwal, 2014, Chen and Suh, 2015]. One of the most popular algorithms proposed in recent times has been the Rank-Centrality algorithm. Negahban et al. [2017] were able to show an upper-bound on the relative $L_2$-error (between true weights and proposed weights) such that the relative $L_2$-error goes to 0 for larger $n$. They did this by finding the stationary distribution of matrix $P$ which was defined as

$$P_{ij} = \begin{cases} \frac{A_{ij}}{d_{max}} & \text{if } i \neq j \\ 1 - \frac{1}{d_{max}} \sum_{k \neq i} A_{ik} & \text{if } i = j \end{cases} \tag{1}$$

Here $A_{ij}$ is the number of times $i$ beats $j$ in $k$ queries divided by $k$, if $(i, j) \in E(G)$ else $A_{ij} = 0$. The pseudocode can be found in Algorithm 1. They also show that their algorithm outperformed other algorithms on various datasets. As our work builds on Negahban et al. [2017] we have included a primer to Rank-Centrality in the Appendix A.1.

---

**Algorithm 1** Rank-Centrality

**Require:** $n$ objects to compare, $E$ the set of edges between these $n$ objects, $A$ the edge weights calculated based on the voter inputs.
**Ensure:** A weighing of the $n$ objects such that the output weight is fairly close to input hidden weights.
 1: Compute the transition matrix P according to Equation 1
 2: Compute the stationary distribution $\pi$ (as the limit of $p_{t+1}^T = p_t^T P$).

---

Besides research on adversarial attacks in ranking, there has also been research on how the robustness of the ranking algorithms when exposed to noise. Cucuringu [2016] proposed Sync-Rank, by viewing the ranking problem as an instance of the group synchronization problem over the group $SO(2)$ of planar rotations. This led to a robust algorithm for ranking from pairwise comparisons, as was verified by synthetic and real-world data. Mohajer and Suh [2016] worked on a stochastic noise model in order to find top-$K$ rankings. They were able to show an upper-bound on the minimum number of samples required for ranking and were able to provide a linear-time algorithm that achieved the lower-bound. However, both of these works [Cucuringu, 2016, Mohajer and Suh, 2016] were primarily tackling the ranking problem when there was noisy and incomplete data as opposed to an adversary who was trying to disrupt the algorithm.

Suh et al. [2017] worked on the adversarial ranking problem considering two cases one where the fraction of adversarial voters is known and is unknown. They got results that were asymptotically equivalent to Chen and Suh [2015]'s Spectral-MLE algorithm despite the presence of adversarial voters. However, they worked with the significantly weaker assumption that the adversarial voters will vote exactly opposite to the faithful voters (i.e. if the good voters voted that $i$ is better than $j$ with probability $\frac{w_i}{w_i + w_j}$ then the Byzantine voters voted with probability $\frac{w_j}{w_i + w_j}$). Our setting is much more general in that we do not make any assumptions regarding the Byzantine voters.

Agarwal et al. [2020] worked on rank aggregation when edges in the graph were compromised. They were able to prove fundamental results regarding the identifiability of weights for both general graphs as well as Erdős-Rényi graphs. They also came up with polynomial-time algorithms for identifying these weights. However, their model is significantly different from ours. They assume that some of the comparisons in the comparison graphs might have been completely compromised while the others are free from any kind of corrtuption. In practice this is unlikely to happen, we assume a stronger model where every edge can have good and Byzantine voters. A detailed comparison with Agarwal et al. [2020] made in section 3.5

## 2 Problem Setting

Given a set of $n$ objects, our goal is the obtain a ranking from a set of pairwise comparisons over the objects. We assume that the pairs to be compared are determined by an Erdős-Rényi comparison graph $G(n, p) = (V, E)$. Each edge in the graph corresponds to a pair that can be compared and we allow at most $k$ comparisons per pair.

The pairs can be compared by voters who fall into one of two categories - good or *Byzantine*. Out of $K$ total voters, $F$ are assumed to be Byzantine. Every good voter when queried to compare a pair of objects, votes stochastically according to a Bradley-Terry-Luce (BTL) preference model parametrized by a weight/score vector $w \in \mathbb{R}^n_+$ where the probability of preferring object $i$ over object $j$ is given by $\frac{w_i}{w_i + w_j}$. The Byzantine voters on the other hand can vote however they wish (stochastic/deterministic). To make the problem general we consider a powerful central-adversary that could potentially control all of the Byzantine voters. We assume that the Byzantine adversary knows (1) The queries asked to the good voters and their corresponding votes (2) The comparison graph $G(n, p)$ (3) The learning algorithm that is used to produce the ranking (4) The underlying true BTL weights of the objects that the good voters use to produce their comparisons.

We assume that both the good voters and the byzantine voters will produce the same output when the same pair is asked to be compared multiple times by them.[4]

The goal of a learning algorithm is to use the least number of voter comparisons to output a score vector that is as close to the true scores. The learning algorithm is allowed to choose any voter to compare any pair of objects. However, it does not know the type of voter - good or Byzantine and naturally, a good algorithm must be robust to the presence of the powerful Byzantine voters. Finally, we work in the passive setting, where a learning algorithm is needed to fix the *pair to voter* mapping apriori before the votes are collected.

## 3 Results for the Erdős-Rényi Graph

We focus our efforts on the Erdős-Rényi graph primarily because it has been heavily studied in the context of rankings. We pay careful attention to the $p \in \Theta(\log n / n)$. We go on to find that our algorithms turned out to be exponential in terms of the degree. This makes the Erdős-Rényi graph an optimal candidate for working out our algorithms because the degrees of the Erdős-Rényi graph will also be logarithmic with high probability, therefore giving us polynomial-time algorithms.

### 3.1 Rank-Centrality Failure

We initially try to motivate the need for a robust version of the Rank-Centrality algorithm by showing that even a simple strategy from a Byzantine adversary will lead to an unsuitable ranking. In this scenario, the Byzantine adversary does not know the comparison graph, nor does it know the votes of good voters, and still can beat the Rank-Centrality algorithm. Formally we show that:

**Theorem 5.** *Given $n$ objects, let the comparison graph be $G(n, p)$. Each pair in $G$ is compared $k$ times by voters chosen uniformly at random amongst the $K$ voters. If all Byzantine voters vote according to the opposite-weight strategy in an Erdős-Rényi graph with $p = C \log n / n$ then there exists a $\tilde{\pi}$ for which the Rank-Centrality algorithm will output a $\pi$ such that*

$$\frac{\|\pi - \tilde{\pi}\|}{\|\tilde{\pi}\|} \geq \frac{C \log n}{(2\sqrt{b} + b + 2bC \log n)} \cdot \frac{b - 1}{8\sqrt{2}b(b + 1)} \cdot \frac{F}{K}$$

*with probability $\geq 1 - \frac{1}{n^{knCF^2/64K^2}} - \frac{1}{n^{C^2 \log n/8}} - \frac{1}{n^{C/3-1}}$. Here $b$ is the skew in object weights i.e. $b = \max_{i,j} w_i / w_j$.*

Here when we say the opposite-weight strategy we mean a strategy where the Byzantine voter will vote for $i$ if $w_i \leq w_j$ otherwise will vote for $j$. If we carefully examine the RHS, we see that the first term asymptotically converges to $\frac{1}{2b}$. The third term is also clearly a constant if there is a constant fraction of Byzantine voters. Therefore, showing the existence of a strategy that will with high

---

[4]This ensures that simple algorithms which query all voters to vote on the same pair multiple times and obtain a median preference will not work.

probability, significantly deviate the weights given by the Rank-Centrality algorithm. The detailed analysis can be found in Appendix A.2.

*Proof Sketch.* We initially show that

$$\frac{\|\pi - \tilde{\pi}\|}{\|\tilde{\pi}\|} \geq \frac{\|\tilde{\pi}^T \Delta\|}{\|\tilde{\pi}\| \cdot (2\sqrt{b} + b\|\Delta\|_2)} \tag{2}$$

Here $\pi$ is the stationary distribution of the transition matrix ($P$) defined in Equation 1 and $\Delta$ denotes the fluctuation of the transition matrix around its mean ($\tilde{P}$), such that $\Delta \equiv P - \tilde{P}$. We initially prove that $\|\Delta\|_2$ will at most be $1 + d_{max}$, since there are at most $1 + d_{max}$ terms in any row or column and each term can deviate by at most 1. Following this, we show that with probability $\geq 1 - \frac{1}{n^{C/3-1}}$ we will have $d_{max} \leq 2C \log n$. Finally, we show that $\|\tilde{\pi}^T \Delta\|$ will be $\in \Omega(pn)$ with probability $\geq 1 - \frac{1}{n^{knCF^2/64K^2}} - \frac{1}{n^{C^2 \log n/8}}$ by constructing a weight-distribution where half the weights are low and half were high. An arbitrary weight-distribution could not be used because if $\tilde{\pi} = \left[\frac{1}{n}, \dots, \frac{1}{n}\right]$ we would get $\tilde{\pi}^T \Delta = 0$ for all $\Delta$. This is because the row sum of $P$ and $\tilde{P}$ will always be 1. $\square$

## 3.2 Byzantine Spectral Ranking

In this section, we propose a novel algorithm called Byzantine Spectral Ranking (BSR). The BSR algorithm relies on asking multiple voters multiple queries and based on the collective response decides whether a voter is acting suspiciously or whether it is plausible that the voter might be a good voter. For each object $i$ to be compared, we take a set of $d_i$ objects to compare it against and have $k$ voters compare $i$ with all of the $d_i$ objects (leading to $kd_i$ comparisons for each voter). Based on the voter's responses, we try to eliminate voters who are likely to be Byzantine. The BSR algorithm ensures that a constant fraction of good voters remain and that if any Byzantine voters remain then these voters have votes similar to good voters. Intuitively, the BSR algorithm goes over every single subset of $[d_i]$ and tries to find if the Byzantine voters have collectively voted strongly in favor or against this subset. The pseudocode can be found in Algorithm 2.

---

**Algorithm 2** Byzantine Spectral Ranking

---

**Require:** $n$ objects, $K$ voters, Comparison graph $G$, $k$ comparisons allowed per edge, parameter $Q$.
**Ensure:** Weights ($\pi$) for each of the objects in $[n]$
 1: **procedure** Bound_Sum_Deviations($i, S, d, k, \alpha$)
 2:     Pick $k$ voters randomly from the voter set ($[1, \dots, K]$)
 3:     The $k$ voters are queried to compare all objects in $S$ with $i$          $\triangleright k \times d$ queries in total
 4:     $T$ represents the binary representation of the voters' inputs          $\triangleright T$ will be $k \times d$ matrix
 5:     $\delta \leftarrow \sqrt{\frac{Q}{2} d \log k}$          $\triangleright$ We show later that $Q$ needs to be $\geq 1$
 6:     $M \leftarrow [0]_{2^d \times k}$
 7:     **for** each $\xi \in \{1, -1\}^d$ **do**
 8:         $U \leftarrow T\xi$
 9:         $\hat{m} \leftarrow \mathsf{Median}(U)$
10:         $M[\xi][v] \leftarrow 1$ if voter $v$ is $5\delta$ away from $\hat{m}$ and 0 otherwise
11:     **end for**
12:     max_out $\leftarrow 8k^{1-Q} + 8k^{1-\alpha}$
13:     Remove voters with $M[\xi][v] = 1$ if the sum of $M[\xi]$ is $\geq$ max_out for some $\xi$
14:     Return leftover votes
15: **end procedure**
16: $\alpha \leftarrow 1 - \log d_{max}/\log k$          $\triangleright \alpha$ is a parameter whose value is set during analysis
17: **for** each object $i$ to be compared **do**
18:     votes $\leftarrow$ Bound_Sum_Deviations($i, N_G(i), d_i, k, \alpha$)
19:     Using votes update $P$ as described in Equation 1
20: **end for**
21: Compute the stationary distribution $\pi$ which is the limit of $p_{t+1}^T = p_t^T P$.

---

### 3.2.1 Analysis

The detailed analysis can be found in Appendix A.3. Based on the proof of the Rank-Centrality Convergence the only result which does not hold for Byzantine voters is Lemma 3, where they bound the 2-norm of the $\Delta$ matrix. We show that:

**Lemma 6.** *For an Erdős-Rényi graph with $p \geq 10C^2 \log n/n$ with $k \in \Omega(np)$ comparisons per edge, algorithm 2 removes some voters such that we have:*

$$\|\Delta\|_2 \leq 80 \max\left(\frac{d_{max}}{k}, \sqrt{\frac{\log k}{d_{min}}}\right)$$

*with probability $\geq 1 - 16n^{(1-1.5C^2)}$.*

*Proof Sketch.* Proving that $\|\Delta\|_2$ is bounded with high probability turns out to be equivalent to proving that $\mathbb{P}(\sum_{j \in \partial i} |C_{ij}| > kd_i s)$ is bounded. $C_{ij}$ is defined as $kA_{ij} - kp_{ij}$, where $p_{ij} = \frac{w_i}{w_i + w_j}$. Since it is a sum of $d_i$ absolute values it can be upper-bounded by considering the union bound over all $2^{d_i}$ combinations for this sum.

We initially show that with high probability for all $2^{d_i}$ values of $\xi$ the mean$(\hat{m})$ that we compute in the BSR algorithm will only be $\mathcal{O}(\delta)$ away from the expected mean. This gives us a good check for whether a voter is acting out of the ordinary or not. The algorithm then filters out voters that are collectively acting out of the ordinary for a particular $\xi$. We show that at the end of these removals, the number of voters will only be a constant factor far from $k$, i.e. most of the good voters will be retained. Finally, we show an upper-bound on the row sum of the absolute values of the deviations of the remaining Byzantine voters to complete the proof. □

Intuitively, we get the terms $d_{max}/k$ and $\sqrt{\log k/d_{min}}$ in Lemma 6 by considering the maximum average deviation for the row sum that can be caused for any $\xi$ by the Byzantine voters. This can be written as max_out voters having a deviation of $d$ and the remaining $k$ voters having a deviation of $\delta$. Therefore we can see that the overall deviation will be $\frac{1}{kd}$ (max_out $\cdot d + \delta \cdot k$). Substituting the values of the parameters, and choosing an appropriate $Q$ we get $\mathcal{O}\left(\max\left(\frac{d_{max}}{k}, \sqrt{\frac{\log k}{d_{min}}}\right)\right)$. Finally, using lemma 6, and a few lemmas from Negahban et al. [2017] we conclude that:

**Theorem 7.** *Given $n$ objects, let the comparison graph be $G(n, p)$. Each pair in $G$ is compared $k$ times with the outcomes of comparisons produced as per the Byzantine-BTL model with weights $[w_1, \ldots, w_n]$. Then there exists an algorithm such that when $F \leq K(1 - \epsilon)/2$, $p \geq 10C^2 \log n/n$, $\epsilon > 0$ and $k \geq 18d_{max}/\epsilon^2$ the following bound on the error*

$$\frac{\|\pi - \tilde{\pi}\|}{\|\tilde{\pi}\|} \leq 480b^{5/2} \max\left(\frac{d_{max}}{k}, \sqrt{\frac{\log k}{d_{min}}}\right)$$

*holds with probability $\geq 1 - 6n^{-C/8} - 16n^{(1-1.5C^2)}$.*

**Remark.** The BSR algorithm gives a suitable ranking in polynomial time, but is fairly slow as the complexity will be $\in \Omega(n^{5C^2+1})$, which is not suitable for larger values of $n$ and $C$.

**Remark.** Our process of filtering out Byzantine voters is tightly tied to the analysis of the Rank-Centrality algorithm and the procedure to remove voters may not be suitable for other algorithms. The advantage of tying the removal procedure to the ranking algorithm is that we would not remove voters who may vote in an adversarial fashion, but do not significantly affect the row sum of the absolute difference between the empirical and true Markov chain transition matrices. An algorithm that tries to explicitly identify Byzantine voters might end up removing such voters too (and hence their votes for other useful pairs) which might be counter-productive.

### 3.3 Fast Byzantine Spectral Ranking

While Algorithm 2 performs asymptotically similar to the Rank-Centrality algorithm, it is fairly slow requiring at least $\Omega(2^{d_{min}} \cdot n)$ time. We now present the Fast Byzantine Spectral Ranking (FBSR) Algorithm for more practical settings. Essentially in our previous algorithm, we ensure

that $\mathbb{P}(\sum_{j \in \partial i} |C_{ij}| > kd_i s)$ is upper-bounded. This takes up a lot of time because essentially we iterate over all $\xi \in \{-1, 1\}^{d_i}$ in the algorithm. But, this can be improved. If $d_i = \beta \log_2 n$ we can bucket the entries into $\beta$ buckets ($[B_1, \ldots B_\beta]$) of size $\log_2 n$ each. Therefore an equivalent formulation would be to upper-bound $\sum_{l=1}^{\beta} \mathbb{P}(\sum_{j \in \partial i \,\&\, j \in B_l} |C_{ij}| > \frac{kd_i s}{\beta})$. Here instead of summing up $d_i$ absolute values, we will only be summing up $\log_2 n$ values $\beta$ times.

---

**Algorithm 3** Fast Byzantine Spectral Ranking

1: $\alpha \leftarrow 1 - \log((2 + C/8)\log n)/\log k$
2: **for** each object $i$ to be compared **do**
3:     max_size $\leftarrow \log_2 n$                                                   $\triangleright$ Can be made larger than $\log_2 n$
4:     $\beta \leftarrow \left\lceil \frac{d_i}{\text{max\_size}} \right\rceil$
5:     Split $N_G(i)$ into $\beta$ buckets such that they are equally sized and let them be $[B_1, \ldots, B_\beta]$
6:     **for** bucket $\leftarrow 1$ to $\beta$ **do**
7:         votes $\leftarrow$ Bound_Sum_Deviations($i$, $B_{\text{bucket}}$, size($B_{\text{bucket}}$), $k$, $\alpha$)
8:         Using votes update $P$ as described in Equation 1
9:     **end for**
10: **end for**
11: Compute the stationary distribution $\pi$ which is the limit of $p_{t+1}^T = p_t^T P$.

---

**Theorem 8.** *Given $n$ objects, let the comparison graph be $G(n, p)$. Each pair in $G$ is compared $k$ times with the outcomes of comparisons produced as per the Byzantine-BTL model with weights $[w_1, \ldots, w_n]$. Then there exists an algorithm such that when $F \leq K(1 - \epsilon)/2$, $p = 10C^2 \log n/n$, $\epsilon > 0$, and $k \geq 18(2 + C/8)\log n/\epsilon^2$ the following bound on the error*

$$\frac{\|\pi - \tilde{\pi}\|}{\|\tilde{\pi}\|} \leq 480b^{5/2} \max\left(\frac{\log n}{k}, \sqrt{\frac{\log \log n}{\log n}}\right)$$

*holds with probability $\geq 1 - (6 + 240C^2)n^{-C/8}$ that runs in $\mathcal{O}(n^2)$ time.*

The detailed analysis can be found in Appendix A.4.

### 3.4 An Impossibility Result and optimality of FBSR

The main idea in the BSR algorithm was the estimation of an accurate mean for all values of $\xi$. However, we can see that finding the mean can become very challenging if the Byzantine voters outnumber the good voters. This is because the Byzantine voters can always create a shifted binomial distribution and trick us into believing that an entirely different value is the mean.

We answer a natural question: is it even possible to find algorithms when $F$ crosses $K/2$? We prove a fundamental result showing that for any ranking algorithm to give a satisfactory result, a majority of good voters will always be required. Formally we show that:

**Theorem 9.** *If $F \geq K/2$, then no algorithm can for all weights ($\tilde{\pi}$), output weights ($\pi^*$) such that*

$$\frac{\|\pi^* - \tilde{\pi}\|}{\|\tilde{\pi}\|} \leq f(n)$$

*with probability $> 1/2$, where $f(n)$ is a function that converges to 0 as $n$ goes to $\infty$.*

*Proof Sketch.* We prove this result by contradiction, let us suppose there is an algorithm $\mathcal{A}$ which can give a satisfactory ranking for all weights. We consider the $F = K/2$ case, as the case where $F > K/2$ can clearly be reduced to the $F = K/2$ case.

We then consider two instances: (1) Good voters vote according to $\tilde{\pi}$ and the Byzantine voters vote according to $\tilde{\pi}'$. (2) Byzantine voters vote according to $\tilde{\pi}$ and the good voters vote according to $\tilde{\pi}'$. We go on to claim that when $F = K/2$ these instances are indistinguishable to any algorithm. Since $\mathcal{A}$ succeeds with probability $> 1/2$ for any instance, we can say by using the union bound that there must be a $\pi^*$ such that it is close to both $\tilde{\pi}$ and $\tilde{\pi}'$. We can then choose the values of $\tilde{\pi}$ and $\tilde{\pi}'$ such that they are far from each other and therefore showing that there is no $\pi^*$ that it is close to both $\tilde{\pi}$ and $\tilde{\pi}'$. Giving us a contradiction. $\square$

The detailed proof can be found in Appendix A.5.

**Remark.** This impossibility consequently shows us that the FBSR algorithm (cf. Theorem 8) is optimal in terms of tolerance towards the Byzantine fraction ($F/K$).

### 3.5  Comparison with Agarwal et al. [2020]

While the setting of Agarwal et al. [2020] is different from our setting as mentioned in Section 1.2, if we were to consider both works in Byzantine voter setting a comparison could be made by considering an edge that has a Byzantine voter to be corrupted as per their model. Here, their algorithm ,to have a valid convergence proof, needs $k \in \Omega(\log n)$. This would imply that $F/K \in O(\log\log n/\log^2 n)$ for the sparse graphs and $F/K \in O(1/\log n)$ for denser graphs. Our algorithms can potentially tolerate a significantly higher corruption rate of $1/2$.

Another way to compare our results with Agarwal et al. [2020] would be to set $k = \Omega(\log n), p = \Theta(\log n/n)$ for our model and $k' = 1, p' = pk/k'$ for their model. This allocation ensures only a single voter per edge for their model and equal number of comparisons in expectation. We see that their algorithm does not converge ($L_1 \in \mathcal{O}(\sqrt{\log n})$), while our algorithms converges $\left(L_2 \in \mathcal{O}\left(\max\left(\frac{\log n}{k}, \sqrt{\frac{\log k}{\log n}}\right)\right)\right)$. Furthermore, our algorithm can handle a corruption rate of $1/2$ while their algorithm can handle a corruption rate of $\mathcal{O}\left(\log\log n/\log n\right)$[5].

## 4  Experimental Results

In this section, we confirm our theoretical bounds with experiments conducted on synthetic and real data. While truly Byzantine strategies might be hard to find and can even be computationally infeasible, we consider some practically possible strategies. We show the performance of Rank-Centrality against these strategies, supporting our results in section 3.1. We then proceed to show that these strategies are not as effective against the BSR/FBSR algorithm which gives better rankings.

**Experimental Settings:** We see that for the FBSR Algorithm to work well we need the existence of at least a few entries that are at a distance $\geq 5\delta$ from $\hat{m}$. However, since we are summing up $\log n$ entries we will need $5\delta \leq \log n$. Otherwise, the BSR algorithm will run exactly as the Rank-Centrality algorithm. Using the value of $\delta$ from Algorithm 3 and setting $Q = 1$ and using $k \geq \log n$ we get:

$$\frac{25}{2}\log\log n \leq \log n$$

While it is asymptotically true, we see that the RHS only becomes greater than the LHS when $n \geq 1.18 \times 10^{21}$. However, as our proof was designed to work against any Byzantine strategy, and at the same time the convergence derivation of Negahban et al. [2017] was fairly loose (for example, the union bound of $2^d$ probabilities to ensure the sum of absolute terms is bounded) the FBSR algorithm empirically works for a variety of strategies even for significantly smaller thresholds. We applied the FBSR Algorithm with the following modifications considering the smaller $k$ and $n$ values: (1) setting $5\delta = 1 + \sqrt{\text{size}(B_{\text{bucket}})}$ and (2) setting max_out as $k/20$. All experimental results in sections 4 and 4 have been taken by taking the mean over 10 random trials.

**Synthetic Data:** We consider the following parameters in our testing (1) $n = 200$ (2) $k = 100$ (3) $p = 20\log n/n$ and (4) $w_i = \text{Uniform}(1, 100)$. These are then normalized such that $\sum \tilde{\pi}_i = 1$. We compare our model with the Rank-Centrality algorithm on synthetic data. We consider four potential strategies used by Byzantine voters:

(1) *Fixed Order Vote:* The Byzantine voters vote according to some random pre-determined order. This is a fairly simple strategy that does not even require the Byzantine voters to know the $\tilde{\pi}$ values.

(2) *Opposite Vote:* The Byzantine voters vote exactly the reverse of the weights. We had shown in section 3.1 that this strategy leads to a failure for some weights.

---

[5]they also come up with an algorithm for denser graphs that can handle a corruption rate of $1/4$, however the algorithm has worse time-complexity and requires $p \in n^{\epsilon-1}$ for $\epsilon > 0$

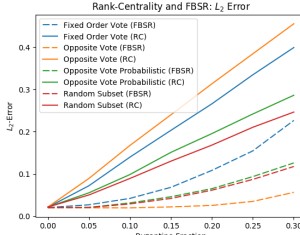
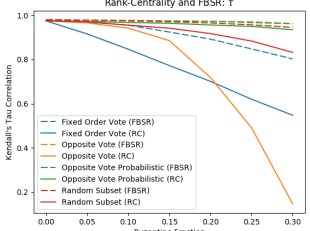
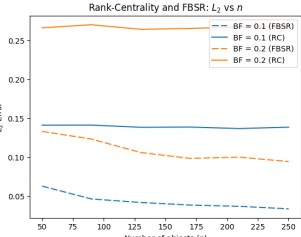

Figure 1: $L_2$ vs $F/K$   Figure 2: $\tau$ vs $F/K$   Figure 3: $L_2$ vs $n$

(3) *Opposite Vote Probabilistic:* The Byzantine voters are equivalent to good voters, except their final answer is flipped. For example, the Byzantine voters will vote for object $j$ with probability $\frac{w_i}{w_j+w_i}$ and vote for object $i$ with probability $\frac{w_j}{w_j+w_i}$.

(4) *Random Subset:* For each pair, the Byzantine voters either decide to behave like good voters with half probability or decide to *o*pposite vote.

Figures 1 and 2 demonstrate the performance of the Rank-Centrality algorithm and the FBSR algorithm against the aforementioned strategies. For Figure 1, we plot the relative $L_2$-Error against the Byzantine Fraction ($F/K$). Similar to our results in section 3.1, we see that for most strategies the relative $L_2$-Error for the Rank-Centrality Algorithm grows linearly with $F/K$. We can see that the relative $L_2$-Error for both Fixed Order Vote and Opposite Vote strategies are fairly high for the Rank-Centrality algorithm and that the Fixed Order Vote dominates the relative $L_2$-Error for the FBSR algorithm. However, even though the Fixed Order Vote is the best strategy against the FBSR algorithm. The FBSR algorithm is still able to perform around two times better than the Rank-Centrality algorithm across all values of $F/K$.

We also confirm that our rankings are close to the actual rankings. To do this we consider Kendall's Tau Correlation [Kendall, 1938] between our ranking and the actual ranking. From Figure 2, we see that for the Rank-Centrality algorithm, the Opposite Vote strategy is able to give very low values of $\tau$. When the Byzantine fraction is around $0.3$ we get a Kendall's Tau Correlation close to $0$ (essentially a random ranking). We also see that the FBSR algorithm is able to give better results for all strategies.

To further show the convergence of the algorithms in the Byzantine-BTL model for larger $n$, we plot the relative $L_2$ error and Kendall's Tau Correlation against the number of objects in Figures 3 and 4 for the Fixed Order Vote strategy. We consider values of $F/K$ in $[0.1, 0.2]$. These results come when we have set $k = n$. We see that for a large range of the number of objects both metrics have remained constant for the Rank-Centrality algorithm thus supporting our hypothesis. On the other hand, we see that as we increase the number of objects the $L_2$-error gradually falls and the Kendall's Tau Correlation gradually increases. The rather slow descent of the relative $L_2$-error is backed by Theorem 8 where we see that for larger $k$ the relative $L_2$ error is $\in \mathcal{O}\left(\sqrt{\frac{\log\log n}{\log n}}\right)$.

**Real Data**: Experimentation with real data was not straightforward, considering that most datasets do not have voter labels nor do they have voter queries as requested by the BSR algorithm (one voter is always asked multiple queries of the form $(i, j)$ where $i$ is fixed and $j$ is varied). To get around this issue, we use complete rankings. We consider the Sushi dataset comprising $5000$ voters ordering the $10$ sushis from most preferred to least preferred. Given the preference order, we get $45$ pairwise votes from each of the voters. Since we are assuming that we are dealing with permutations here, we assume the Byzantine voters can use the following strategies:

(1) *Fixed Order Vote / Opposite Vote:* Same as previously defined, but this time the entire permutation is given.

(2) *Opposite Random Flips:* Each Byzantine voter starts with the opposite permutation and then swaps some of the objects at random.

Since the Sushi dataset has many voters we do not use our (2)nd modification mentioned above. We compute the true weights by applying the Rank-Centrality algorithm directly (thus the Rank-Centrality algorithm has a big advantage, especially for smaller weights). Figures 5 and 6 show the Rank-Centrality algorithm and the BSR algorithm's performance. We see that the BSR algorithm achieves

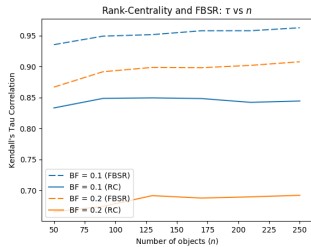
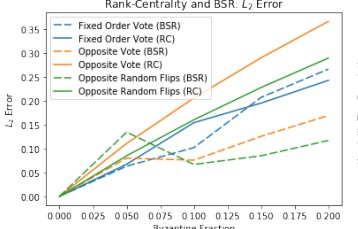
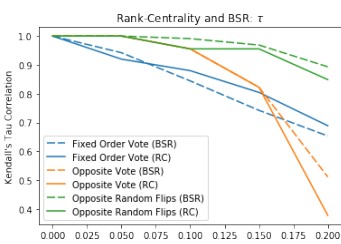

Figure 4: $\tau$ vs $n$        Figure 5: Sushi: $L_2$        Figure 6: Sushi: $\tau$

lower relative $L_2$-errors/Kendall's Tau Correlation for most Byzantine fractions and strategies. The comparatively small improvement in Kendall's Tau Correlation is attributed to a rather small difference in the weights of the various Sushi (5 sushis have weights $\in [0.085, 0.13]$).

## 5   Conclusion and Further Work

We studied the potential disruption that Byzantine voters can do to the widely-used Rank-Centrality algorithm and proposed fast solutions to counter the same. Our algorithms and analysis techniques are quite non-trivial and indicate to us the need for developing this theory further. We require $d \in \Theta(\log n)$ and $k \in \Omega(\log n)$ for the success of the BSR and FBSR algorithms. However, Negahban et al. [2017] only required $d \in \Omega(\log n)$ and $k \in \Omega(1)$. An interesting direction would be to explore whether sub-logarithmic values of $k$ will be possible. Alternatively, a lower bound on the number of required votes per edge for an algorithm to succeed with constant probability would also be an interesting direction.

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
