# OpenReview forum: "Byzantine Spectral Ranking"
_NeurIPS.cc/2022/Conference — NeurIPS 2022 Accept_

### Official Review · Reviewer_gNNK · 2022-07-06

**Rating:** 7
**Confidence:** 4
**Soundness:** 3 good
**Presentation:** 2 fair
**Contribution:** 3 good

**Summary:**

This paper tackles the problem of learning from comparison labels in crowd-sourced settings, where multiple labelers can label a pair and some labelers may be spammers/Byzantine unlike standard/good labelers. It is assumed that the comparisons are generated by the The Bradley-Terry (BT) model and form an Erdos-Renyi graph. Main contributions begin with theoretical results showing that the state-of-the-art Rank Centrality (RC) algorithm cannot generalize its error bounds for BT model estimation from Byzantine comparison labels. Motivated by this, this paper proposes a poly-time Byzantine Spectral Ranking (BSR) algorithm that first identifies potential comparisons from Byzantine labelers by searching over the neighborhood of each node in the comparison graph and identifying labelers that significantly deviate from the median score estimate, and then applies the RC algorithm after removing Byzantine comparisons. Theoretical results show that if strictly less than half of the labelers are Byzantine, removing Byzantine comparisons via BSR improves over vanilla RC estimates with high probability. This observation is validated via experiments over a synthetic dataset and the benchmark Sushi dataset, where comparisons are generated from total rankings, using l_2 error and Kendall-Tau correlation metrics.

**Questions:**

Melnyk et al., Byzantine Preferential Voting (https://doi.org/10.1007/978-3-030-04612-5_22) seems to have a similar problem setting. Can the authors summarize and contrast the differences with the current paper?

**Limitations:**

Limitations regarding sample complexity are discussed in the conclusions, and there are no potential negative societal impacts.

**Strengths And Weaknesses:**

Strengths: The problem of learning from corrupted comparison labels is interesting and significant. The spectral ranking algorithm and following theoretical/experimental results in this setting are novel.

Weaknesses: Introduction is very dense with a lot of technical notation; more motivation and insights rather than technical descriptions would be helpful.

Notation is inconsistent (e.g., w or \pi or p for scores, v or i for labeler index in Algorithm 1), with some used without introduction (e.g., N_g(i) in Algorithm 1). The value of parameter alpha is never introduced.

Supplemental material is not referenced in the main paper, though all proofs and extra experiments are there.

In practice, a passive-learning algorithm that can use the comparison label of any pair from any labeler would require total rankings from each labeler, which are costly to collect. Meanwhile, the authors recognize this practical limitation.

Experiments would be stronger if the convergence time of BSR vs. RC is reported to observe the prediction performance vs. time trade-off.

---

> ### Author Response · Authors · 2022-08-02
> **Addressing the concerns of Reviewer gNNK**
>
> Thank you for the positive review and useful feedback. Thank you also for recognizing the novelty of the problem and our results.
> >Introduction is very dense with a lot of technical notation.
>
> We will expand into the motivation and insights while deferring some of the technical notation to the results section.
> >Notation is inconsistent (e.g., $w$ or $\pi$ or $p$ for scores, $v$ or $i$ for labeler index in Algorithm 1), with some used without introduction (e.g., $N_g(i)$ in Algorithm 1). The value of parameter alpha is never introduced.
>
> 1. We were following the weights notation used in the Rank Centrality paper. Specifically, $w$ are the latent weights, $\pi$ are the weights normalized to $1$, and $p_t$ are the weights in the $t^{th}$ iteration of the algorithm.
> 2. We will change the $i$ in line 10 in Algorithm $1$ to $v$ and introduce the $N_{G}(i)$ notation as the neighbors of $i$ in $G$.
> 3. The value of $\alpha$ was introduced on line 16 in Algorithm 1 and on line 1 in Algorithm 2 and is set during the analysis. However, we will add a line clarifying the same in our work.
>
> >Supplemental material is not referenced in the main paper
>
> We will add the appropriate references to the Appendix in the paper. We will also create a repository and provide a link for the same in the paper.
> >Experiments would be stronger if the convergence time of BSR vs. RC is reported to observe the prediction performance vs. time trade-off.
>
> For our setting of (1) $n = 200$, (2) $k = 100$ , and (3) $p = 20\log n/n$, we summarise the average time and performance of both the $L_2$-error and the Kendall-Tau Correlation in Tables 1 and 2.
> |Algo|Bucket|Time(s)|0|0.05|0.1|0.15|0.2|0.25|0.3|
> |-|-|-|-|-|-|-|-|-|-|
> |RC||1|0.022|0.072|0.14|0.203|0.266|0.335|0.399|
> |FBSR|6|24|**0.019**|0.042|0.085|0.143|0.196|0.272|0.372|
> |FBSR|9|185|0.029|0.04|0.067|0.107|0.164|0.228|0.312|
> |FBSR|12|1034|0.02|**0.026**|**0.039**|**0.062**|**0.097**|**0.139**|**0.204**|
>
> Table 1: Bucket size, Time, and $L_2$ errors
> |Algo|Bucket|0|0.05|0.1|0.15|0.2|0.25|0.3|
> |-|--|-|-|-|-|-|-|-|
> |RC||0.974|0.916|0.846|0.773|0.701|0.62|0.548|
> |FBSR|6|**0.98**|0.951|0.911|0.846|0.807|0.747|0.659|
> |FBSR|9|0.975|0.961|0.934|0.889|0.848|0.792|0.756|
> |FBSR|12|0.978|**0.971**|**0.96**|**0.934**|**0.906**|**0.87**|**0.834**|
>
> Table 2: Bucket size and Kendall-Tau Correlation
>
> As we increase the bucket size, the performance improves for the FBSR Algorithm with a corresponding increase in time. The FBSR algorithm manages to outperform the RC algorithm even for small values of bucket size. Furthermore, as expected the time taken by the FBSR algorithm is exponential in the bucket size. We also provide a plot showing how the performance varies for different bucket sizes in the Supplementary Material in Rebuttal/performance.jpg.
> >Melnyk et al., Byzantine Preferential Voting seems to have a similar problem setting. Can the authors summarize and contrast the differences with the current paper?
>
> Thanks for bringing this work to our attention. There are several differences between our setting and the setting considered by Melnyk et al and we summarize the same below.
>
> While both papers address the question of obtaining rankings when the data is gathered from a crowd-sourced setting with some of the votes coming from malicious/adversarial sources, the key difference is that each party (voter) provides a complete ranking in Melnyk et al while the voters only provide pairwise comparisons in our work. Even if we were to convert the rankings into comparisons, the number of comparisons done by every voter in Melnyk et al. is $O(n^2)$ which may be prohibitively large.
>
> Furthermore, we work in a stochastic setting and our algorithms are robust to the implicit randomness associated with the *good* voters while Melnyk et al. work in a deterministic setting with no randomness for the good voters. It is unclear if the results of Melnyk et al. extend easily when the voters provide stochastic rankings generated according to certain standard models of permutations such as the Mallows model. It is known that even with only good voters, finding the Kemeny consensus ranking for the Mallows model is  NP-Hard[1]. Melnyk et al. consider approximations in the deterministic setting. However, in our setting, due to the nature of the Bradley-Terry-Luce model, we obtain an estimate of the true ranking.
>
> Finally, our work derives an upper-bound on the relative $L_2$-error on the weights while their algorithm provides an $\alpha$-approximation to the Kemeny Median ranking(ranking with the least $\sum\tau$ across all input rankings). Our algorithm can estimate the true ranking with a corruption rate of $1/2$, while their algorithm approximates the ranking and can handle a corruption rate of $1/3$.
>
> We believe that due to the above differences, a direct comparison between the two papers is not appropriate.
>
> *[1] Young, H. P. Optimal ranking and choice from pairwise comparisons. Information pooling and group decision making.*

---

> > ### Comment · Reviewer_gNNK · 2022-08-09
> > **Response to rebuttal**
> >
> > Author response was comprehensive, I have raised the rating to 7.

---

### Official Review · Reviewer_hBwN · 2022-07-12

**Rating:** 6
**Confidence:** 4
**Soundness:** 3 good
**Presentation:** 3 good
**Contribution:** 3 good

**Summary:**

This paper studies parameter inference in the Bradley-Terry pairwise comparison model in the setting where a fraction of comparison outcomes can be corrupted adversarially. The theoretical results presented in the paper can be summarized as follows.

1. The popular rank-centrality algorithm is not robust to adversarial corruptions.
2. There exists a quadratic-time algorithm that is robust to adversarial corruptions: it recovers the true parameters with vanishing error in the presence of a constant fraction of adversarial voters, under suitable assumptions
3. No algorithm can recover the true parameters if 50% or more voters are adversarial.

The authors illustrate the results with simulations under various adversarial models.

**Questions:**

1. Comparing the informal theorems 1 & 3 seems to imply that the algorithm presented in this paper converges faster that rank centrality ($1/\sqrt{k}$ vs. $1/k$). Can the authors comment on that?
2. One way to relate the results in this paper to those of Agarwal et al would be to consider every edge that has at least one byzantine voter to be corrupted (In the sense of Agarwal et al.). In that case, how do respective results compare?

Another way to meaningfully compare to Agarwal et al. (in the G(n,p) case) is to set:

- $k = \ell$, $p = c \log n / n$ in the results of this paper,
- $k = 1$, $p = \ell c \log n / n$ in the results of Agarwal et al,

and let $\ell$ grow. In that case:

- in the latter setting, the corruption model of Agarwal et al. matches that of this paper (since there is only one voter per edge)
- the number of comparisons is identical (in expectation), but they are distributed differently (fewer edges but with more voters vs more edges with fewer voters).

How do the results compare in that case?

**Limitations:**

Overall, limitations are discussed adequately, although that discussion could be improved by contrasting the results of this paper to those of Agarwal et al., as mentioned above.

**Strengths And Weaknesses:**

Strengths:

- The question addressed in this work is important and of interest to the NeurIPS community. The setting studied is realistic, and might be particularly relevant to applications in crowdsourcing.
- The theory is comprehensive and addresses several important questions:
    - how does rank centrality cope with adversarial inputs?
    - how can we do better?
    - what is the limit?
- The proposed "fast" variant of the algorithm has a practical running time in the regime of interest in this paper ($p = \Theta(\log n / n)$).
- Overall, the contributions are well explained, the developments are rigorous, and the authors provide useful insights into the problem. The simulation results are helpful in illustrating the theory.

In my opinion, the main weakness of this paper is that it does not contrast its results to those in Agarwal et al. (2020), who address very similar questions under a slightly different corruption model. The authors acknowledge that paper in the last paragraph of the introduction, but never come back to it afterwards.

I believe that the contributions in this paper might be novel and complementary to those of Agarwal et al., but I think they are similar enough to warrant a comparison, both in terms of theoretical results and in empirical simulations. For example, carefully identifying regimes where either approach gives better bounds, and running simulations that are comparable to those in Section 4 of Agarwal et al.

---

> ### Author Response · Authors · 2022-08-02
> **Addressing the concerns of Reviewer hBwN**
>
> Thank you for the review and useful feedback. We would also like to thank you for recognizing our contributions and insights. Regarding the questions please find our responses below:
>
> > I believe that the contributions in this paper might be novel and complementary to those of Agarwal et al., but I think they are similar enough to warrant a comparison
>
> We note below that by using any comparison methods described in the review our results are usually able to surpass those of Agarwal et al. in settings with malicious voters.
>
> > One way to relate the results in this paper to those of Agarwal et al would be to consider every edge that has at least one byzantine voter to be corrupted (In the sense of Agarwal et al.). In that case, how do respective results compare?
>
> If we need a constant fraction of the edges to not be corrupted this would require the Byzantine fraction to be $\in O\left(\frac{1}{k}\right)$. It is worth noting that most of Agarwal et al.'s results are derived when $k \rightarrow \infty$, which essentially implies that for some edges to not be corrupted we would need $F/K \rightarrow 0$.
>
> When it comes to Agarwal et al.'s algorithm we see that for the algorithm to have a valid convergence proof we need $k \in \Omega(\log n)$. This would imply that $F / K \in O( \log \log n / \log^2 n)$ for the sparse graphs and $F /K \in O( 1 /\log n)$ for denser graphs. Our algorithms (BSR and FBSR) can potentially tolerate a significantly higher corruption rate of $1/2$.
>
> >Another way to meaningfully compare to Agarwal et al. (in the G(n,p) case) is to set:
> $k = l, p = c \log n$  in the results of this paper,
> $k = 1, p = cl \log n$  in the results of Agarwal et al,
>
> This comparison might give an *unfair* advantage to our work. Primarily because if we use $k = 1$, Agarwal et al.'s algorithm would not converge as the error bound would come out to $O(\sqrt{\log n})$. Meanwhile by setting $l = \Omega(\log n)$, our algorithms(both the BSR and the FBSR algorithms) will converge to a zero error for larger $n$.
>
> However, a possible comparison would be as follows:
> 1. When the graph is **sparse**: Agarwal et al.'s algorithm can handle a corruption rate of $O(\log \log n / \log n)$, while the FBSR algorithm can handle a corruption rate upto $1/2$. Both algorithms run in $O(n^2)$ time.
> 2.  When the graph is **dense**: Agarwal et al.'s algorithm can handle a corruption factor upto $1/4$ but their runtime no longer remains quadratic. While we never explicitly come up with results for dense graphs it should be noted for denser Erdos-Renyi graphs with the probability of edge formation as $p'$ we can simply delete some of the edges in this graph with probability $1 - \frac{c\log n}{np'}$ to get a sparse Erdos-Renyi graph with $p = c\log n / n$.
> 3. Agarwal et al. show a bound on relative $L_1$-error bound of $O\left(\sqrt{\log n/ k} \right)$ whereas we show a bound on the relative $L_2$-error bound of $O \left( \max\left(\frac{d_{\max}}{k}, \sqrt{\frac{\log k}{d_{\min}}} \right)\right)$. While both bounds converge to 0 for larger values of $n$ we see that Agarwal et al.'s bound converges faster. However, we see that for practical Byzantine strategies our algorithm converges with the rate of $\sqrt{\log n / k}$. The same can be seen in Rebuttal/fast\_conv.png in Supplementary Material in and in Table 1 given below.
>
> |F/K|Metric|50|90|130|170|210|250|
> |-|-|-|-|-|-|-|-|
> |0.1|$L_2$|0.063|0.046|0.042|0.039|0.037|0.034|
> |0.1|$L_2\cdot\sqrt{n/\log n}$|0.114|0.098|0.098|0.098|0.1|0.096|
> |0.2|$L_2$|0.133|0.123|0.106|0.099|0.1|0.094|
> |0.2|$L_2\cdot\sqrt{n/\log n}$|0.241|0.26|0.249|0.25|0.271|0.27|
>
> Table 1: $L_2$ and $L_2\cdot\sqrt{n/\log n}$ as $n$ is varied.
> > Comparing the informal theorems 1 \& 3 seems to imply that the algorithm presented in this paper converges faster that rank centrality ( $1 / \sqrt{k}$ vs. $1 / k$ ). Can the authors comment on that?
>
> In Informal Theorem 3, for the sake of brevity and ease of understanding, we state the theorem for a range of values of $k$. However, if Theorem 7 is considered we see that the precise theorem bounds it by $\max\left(\frac{d_{\max}}{k}, \sqrt{\frac{\log k}{d_{\min}}} \right)$.

---

> > ### Comment · Reviewer_hBwN · 2022-08-08
> > **Response to the rebuttal**
> >
> > Thank you for your detailed response, which addresses my questions convincingly. I strongly encourage you to include a similar discussion contrasting your results to those of Agarwal et al in a future version of the paper.

---

> > > ### Author Response · Authors · 2022-08-09
> > > **Thank you**
> > >
> > > We thank the reviewer for taking time out to respond to our rebuttal. We are glad that the queries of the reviewer have been addressed convincingly. We will be happy to add a discussion comparing the results of our work with that of Agarwal et. al in the final version.

---

### Official Review · Reviewer_R2Qw · 2022-07-16

**Rating:** 5
**Confidence:** 3
**Soundness:** 3 good
**Presentation:** 3 good
**Contribution:** 2 fair

**Summary:**

This paper considers the problem of learning the parameters of the Bradley-Terry-Luce (BTL) model by aggregating pairwise comparisons from different voters. The paper considers a setting where some of the voters can be \emph{Byzantine}, i.e. they can provide adversarial votes in order to force large errors in parameter estimation. The paper shows that the popular Rank Centrality algorithm can incur large error in estimation even if a small constant fraction of the voters are Byzantine. The paper then gives an algorithm that incurs low estimation error when the number of \emph{Byzantine} voters strictly less than half of the total voters, the comparison graph is Erdos-Renyi with p = O(\log(n)/n), and number of votes per pair is \Omega(\log n).

**Questions:**

Besides computational efficiency, is there any other issue that prevents this algorithm from being extended to graphs with larger degrees and non-ER graphs?

**Limitations:**

Since the algorithm needs to know the index of each voter and how they vote on various tasks, it can be challenging to use this algorithm in a setting where privacy is a concern as the algorithm can perhaps learn about voter identity based on their pairwise preferences.

**Strengths And Weaknesses:**

Strengths: The problem considered in this paper is well-motivated. In terms of technical novelty: even though the idea to filter malicious samples/votes based on median is quite well-known, I think that the idea of combining votes from several pairs and taking median over the combination is new.

Weaknesses:

1. The error bound is very weak as it is the max of two terms: (1) d_\max/k and (2) \sqrt{log(k)/d_\min}. The second term seems problematic as the error seems to increase if we increase the # votes per pair and the best error rate possible is \sqrt{\log \log n/\log n}. This can be much worse as compared to the bounds known for the non-adversarial setting. The informal theorem on page 2 (Theorem 3) is misleading as it hides the dependence on this \sqrt{log(k)/d_\min} term and does not mention the restrictions on k, comparison graph etc.

2. The algorithm crucially depends on assigning the tasks in a particular way that ensures that all k voters share d specific tasks. This might not be a very good assumption in practice as there might be several limitations such as lack of expertise that does not allow voters to share so many tasks.

3. The algorithm only works for Erdos-Renyi graphs. It seems unlikely that the ideas in the paper will apply to more general graphs as the algorithm becomes computationally inefficient even if there is one vertex with a high degree.

Overall I think that the paper has a good set of results, but falls short of making a significant impact for this problem due to the weakness of the results and strict assumptions on the comparison graph.

---

> ### Author Response · Authors · 2022-08-02
> **Addressing the concerns of Reviewer R2Qw**
>
> We thank the reviewer for their insightful comments. We believe that we solve a critical and demanding problem - to develop suitable algorithms when the Byzantine adversary has complete freedom and show when such algorithms are not possible. Furthermore showing that the existing Rank-Centrality algorithm which is a widely-used algorithm will fail with high probability under a well-coordinated Byzantine attack showing a serious vulnerability in ranking algorithms.
>
> We hope that through the rebuttal and discussion, we will convince the reviewer to reconsider their evaluation - we believe they have underestimated the technical novelty and difficulty of our results.
> >The error bound is very weak as it is the max of two terms: (1) $d_{\max}/k$ and (2) $\sqrt{\log k/d_{\min}}$.
>
> We agree that this bound is weaker than the bounds in the non-adversarial setting. This is because of the substantially powerful model for the adversary. However, we would also like to mention that the algorithm itself performs reasonably well for most practical adversarial settings. For example, if we consider Table 3, we see that the FBSR converges at a rate of $\sim\sqrt{\log n/n}$. The table for the same can be found below. The graph depicting the same can be found in Supplementary Material in Rebuttal/fast\_conv.png
> |F/K|Metric|50|90|130|170|210|250|
> |-|-|-|-|-|-|-|-|
> |0.1|$L_2$|0.063|0.046|0.042|0.039|0.037|0.034|
> |0.1|$L_2\cdot\sqrt{n/\log n}$|0.114|0.098|0.098|0.098|0.1|0.096|
> |0.2|$L_2$|0.133|0.123|0.106|0.099|0.1|0.094|
> |0.2|$L_2\cdot\sqrt{n/\log n}$|0.241|0.26|0.249|0.25|0.271|0.27|
>
> Furthermore, it should be noted that in most *practical* crowd-sourced settings $k$ is fairly small and would be dictated by a budget available to the planner. When we have poly-logarithmic values of $k$, our results are comparable to the non-adversarial setting.
> >The algorithm only works for Erdos-Renyi graphs. It seems unlikely that the ideas in the paper will apply to more general graphs
>
> We try to argue why our choice for the Erdos-Renyi graphs and particularly the sparse Erdos-Renyi graph is justified below:
> 1. Our problem is modeled in a crowd-sourced setting, where we required multiple voters to give votes corresponding to an item pair that we desire. Perhaps, the reviewer is envisioning a model wherein the comparison graph $G$ is fixed and we can only query voters along an edge pair in $G$. However, we are unable to imagine application scenarios where we have the flexibility to assign the voters the pairs to compare while not having the flexibility to choose the edge pairs.
> 2. We also consider a situation where the comparison graph is specifically provided. A simple modification allows us to use our algorithms for a much larger class of graphs. We can see that any graph $G$ that has a subgraph $H$ such that $\xi(H)$ is a constant and $H$ is near-regular ($d_{\max} \in O(d_{\min})$), can also be used. We do not use the edges in $G - H$ and run our algorithm only on $H$. We can do this because in our analysis we have never assumed that the input graph is random. A suitable example would be a higher-degree Erdos-Renyi graph constructed with probability $p'$. We see that by deleting the edges of $G$ with probability $1 - p/p'$, we will get a sparse Erdos-Renyi graph upon which we can apply our results.
> 3. While we cannot apply the BSR Algorithm directly on a general graph without incurring an exponential complexity, the same is not true for the FBSR Algorithm. The FBSR gives us an $O(n^2)$ algorithm with the same $L_2$ bound. As mentioned above, we simply chose a value of $p$ such that the analysis was convenient.
> 4. Finally, in all prior works on adversarial ranking from pairwise comparisons [1,2] that we know of, the authors have chosen an Erdos-Renyi graph. Again in all these cases (and our setting), the Erdos-Renyi graph is a design choice.
>
> >Is there any other issue that prevents this algorithm from being extended to graphs with larger degrees and non-ER graphs?
>
> As explained above our techniques can be applied for denser graphs as well, with no increased computation.
> >It can be challenging to use this algorithm in a setting where privacy is a concern as the algorithm can perhaps learn about voter identity based on their pairwise preferences.
>
> While we recognize this limitation, it is beyond the scope of this work and is not specific only to this work. In particular, even for the standard rank centrality algorithm, there is no privacy-preserving variant as far as we know. A careful investigation of the privacy-utility trade-off via a statistical notion of privacy such as differential privacy can be done for both the rank centrality and the algorithms proposed in this paper. This is an interesting direction for future work.
>
>
> *[1]\: A. Agarwal, S. Agarwal, S. Khanna, and P. Patil. Rank aggregation from pairwise comparisons in the presence of adversarial corruptions.*
>
> *[2]\: C. Suh, V. Y. F. Tan, and R. Zhao. Adversarial top- k ranking.*

---

### Official Review · Reviewer_xZhK · 2022-07-18

**Rating:** 7
**Confidence:** 3
**Soundness:** 3 good
**Presentation:** 4 excellent
**Contribution:** 3 good

**Summary:**

This paper investigates rank aggregation under the BTL model in an adversarial setting, where a fraction of voters are adversarial and always give pairwise comparisons that are opposite to the ground truth. The authors formally demonstrate that Rank-Centrality, a popular rank aggregation algorithm, can have significantly degraded performance with a small fraction of adversarial voters, assuming an Erdos-Renyi comparison graph. The authors also improve the Rank-Centrality method by proposing an algorithm that detects and removes the adversarial voters. The effectiveness of the proposed method is also empirically verified on synthetic and real data.

**Questions:**

Please see the weaknesses.

**Limitations:**

Yes.

**Strengths And Weaknesses:**

Strengths:
- This paper investigates an interesting and practical problem, ranking aggregation in adversarial settings.
- This paper provides formal analyses of the adversarial vulnerability of a popular ranking aggregation method with clear intuitions.
- This paper proposes a simple yet effective fix for Rank-Centrality in the assumed adversarial setting.
- This paper is clearly written and easy to follow.

Weaknesses:
- The adversarial setting assumes that all the adversarial voters know the ground truth pairwise comparisons and ALWAYS vote for the opposite. This setting is somewhat unrealistic.
- The proposed algorithm seems to rely on the above assumption to detect adversarial voters, which undermines the practical applicability of the proposed algorithm.
- In the case where there are a large fraction of adversarial voters, if the above assumption is assumed, can one directly derive a good estimate of the ranking by inverting all the observed pairwise rankings?

---

> ### Author Response · Authors · 2022-08-02
> **Addressing the concerns of Reviewer xZhK**
>
> Thank you for your positive review. We would also like to thank you for recognizing the practicality of the problem and our approach to solving it.
>
> Please find below our response to the weaknesses/questions.
>
> > The adversarial setting assumes that all the adversarial voters know the ground truth pairwise comparisons and ALWAYS vote for the opposite. This setting is somewhat unrealistic.
>
> We believe there has been a misunderstanding. We wish to highlight that the Byzantine voters are assumed to be significantly more powerful than described by the reviewer. In particular, the Byzantine voters can vote in *any* way they want to vote and not just opposite of the true order as incorrectly assumed by the reviewer. Furthermore, the Byzantine voters can even collude with each other to devise a strategy against the algorithm. In spite of this flexible model for the adversary, our algorithm is able to find the true ranking up to a reasonable error bound. The Byzantine voters knowing about the true weights and the comparison graph only increases their abilities and the potential disruption they can cause to the algorithm.
>
> In our experimental section, we create multiple Byzantine voters and test that our algorithm works against a host of potential voting strategies that the Byzantine voters can employ. One such strategy we have considered is the opposite voting strategy.
>
> > In the case where there are a large fraction of adversarial voters, if the above assumption is assumed, can one directly derive a good estimate of the ranking by inverting all the observed pairwise rankings?
>
> As explained in the response to the previous point, the Byzantine voters can vote in any way they wish and so the algorithm flipping the votes might have no effect on them. An example would be for the Byzantine voters to vote randomly with a probability $= 1/2$ independent of other voters. Our impossibility result is when we consider the more powerful Byzantine voters described earlier.

---

> > ### Comment · Reviewer_xZhK · 2022-08-08
> > **Rating updated.**
> >
> > Thanks for the clarification. I've raised the rating to 7.

---

> > > ### Author Response · Authors · 2022-08-09
> > > **Thank you**
> > >
> > > We thank the reviewer for responding to our rebuttal and also raising the rating.

---

### Meta-Review · Area_Chair_SLPK · 2022-08-23

**Recommendation:** Accept
**Confidence:** Certain

**Metareview:**

This is a nice paper that deals with byzantine corruption in the BTL model. It  first shows that rank centrality performs quite poorly (btw  how does the maximum likelihood estimator, which is known to be minimax optimal in the non-adversarial setting, perform in the Byzantine setting?). The paper then presents algorithms for ranking when the majority of workers are non-adversarial (which is unsurprisingly necessary for any algorithm). We all agree that this paper should be accepted, although different reviewers have different levels of excitement about the paper. A persistent concern seems to be absence of proper comparison with Agarwal et al. and weak-ish bounds. Nevertheless, this paper is above bar and I recommend acceptance.



**Award:**

No

---

### Decision · Program_Chairs · 2022-09-14

Accept